# WS-GRPO: WEAKLY-SUPERVISED GROUP-RELATIVE POLICY OPTIMIZATION

## ABSTRACT

Group-Relative Policy Optimization (GRPO) has emerged as an effective approach for training language models on complex reasoning tasks by normalizing rewards within groups of rollouts. However, GRPO's group-relative advantage estimation critically depends on dense step-wise reward signals throughout the reasoning process. In practice, obtaining such dense supervision requires expensive human annotations of intermediate reasoning steps or carefully designed step-wise reward functions. This creates a significant challenge specific to group-relative methods: while GRPO performs best with dense intermediate feedback, real-world scenarios often provide only sparse outcome supervision—such as final answer correctness or binary trajectory labels. We propose Weakly-Supervised Group-Relative Policy Optimization (WS-GRPO), which addresses this unique limitation by learning to extract dense preference signals from sparse outcome supervision while preserving GRPO's group-relative normalization benefits. WS-GRPO operates in two phases: first, it trains a preference model to distinguish between successful and unsuccessful reasoning patterns using only trajectory-level outcomes; second, it leverages this learned preference model to provide step-wise weakly-supervised rewards that are combined with sparse terminal rewards during group-relative policy optimization. By treating consecutive partial trajectories as preference pairs, our method generates dense feedback signals that complement GRPO's group normalization mechanism without requiring step-by-step human annotations. Theoretically, we provide comprehensive guarantees for WS-GRPO establishing preference model consistency under trajectory-level supervision, policy robustness to preference errors with controllable degradation rates, and generalization bounds that decompose error sources across policy learning, preference modeling, and their interaction. Our experiments on reasoning benchmarks demonstrate that WS-GRPO achieves competitive performance using only weak supervision, making group-relative policy optimization practical when detailed process supervision is limited.

## 1 INTRODUCTION

Large language models (LLMs) are emerging as general-purpose reasoning systems, with applications ranging from mathematical problem solving to scientific discovery. A central challenge is how to reliably optimize these models so that their generated reasoning trajectories lead to correct and coherent outcomes Bommasani (2021); Weidinger et al. (2021). Prior works rely on Reinforcement Learning from Human Feedback (RLHF) Ouyang et al. (2022), yet face fundamental obstacles including limited outcome signals such as final answer correctness and expensive intermediate annotations Bai et al. (2022); Liu et al. (2023).

More recently, Group-Relative Policy Optimization (GRPO) has emerged as a promising direction. By replacing a learned value function with group-normalized advantages, GRPO stabilizes training and improves sample efficiency Shao et al. (2024). This design yields strong sample efficiency and memory savings. However, GRPO's performs best when integrated with informative reward signals throughout the reasoning process Li et al. (2025); Tan & Pan (2025); Fei et al. (2025). In practice, most real-world scenarios provide only weak supervision—such as binary correctness of the final answer—rather than dense step-level feedback Yuan et al. (2024); Cui et al. (2025). The gap between

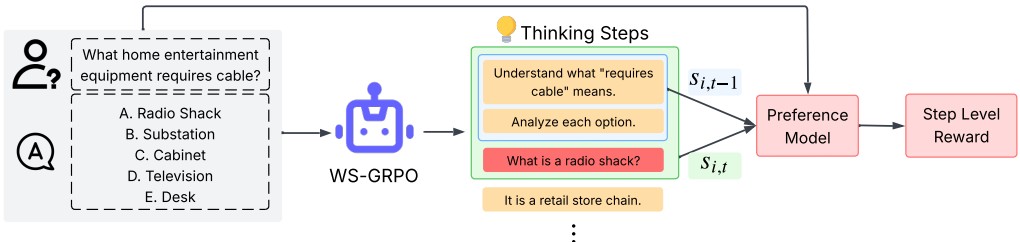

Figure 1: **Weakly-Supervised Step-Level Reward Generation:** WS-GRPO employs a preference model trained on trajectory-level outcomes to generate dense step-wise feedback signals. The model evaluates reasoning progress by comparing consecutive partial trajectories of different lengths. For instance, to assign a reward to step $t$ (e.g., step 3: "What is a radio shack?"), the preference model compares steps up to $(t-1)$ against the extended trajectory containing steps up to $t$, producing a preference score that quantifies the incremental contribution of step $t$. This transforms sparse outcome supervision into dense process-level rewards without step-by-step human annotations.

GRPO's reliance on rich intermediate reward and limited supervision signal in reality remains to be addressed.

Recent work has explored weak supervision as a scalable alternative to explicit process labels. For example, Yuan et al. Yuan et al. (2024) introduce free process rewards derived from outcome labels, while Cui et al. (2025) propose implicit token-level rewards inferred from reasoning traces. Other approaches refine noisy outcome-based labels into denser reward signals Huang et al. (2025); Yi et al. (2025), showing that weak but abundant labels can bootstrap effective reward models. These work collectively show that weak supervision

In this work, we bring the merits from weak supervision to GRPO by proposing Weakly-Supervised Group-Relative Policy Optimization (WS-GRPO). It addresses the challenges by augmenting GRPO with intermediate rewards derived from a weakly supervised preference model (See Figure 1). Specifically, our method operates in two phases: first, a preference model is trained to distinguish between positive and negative reasoning trajectories; second, the trained preference model is reused to generate step-level rewards that complement correctness signals during GRPO optimization. Thus, WS-GRPO provides dense feedback without requiring costly process annotations. By combining weak supervision with group-relative optimization, our approach makes it possible to train reasoning-capable language models under practical constraints. It offers a scalable path toward LLMs in real-world tasks.

Our contributions are as follows:

- We establish comprehensive theoretical guarantees for weakly-supervised group-relative policy optimization through three fundamental results: preference model consistency bounds with optimal convergence rates, policy robustness bounds with controllable error propagation, and generalization bounds decomposing multiple uncertainty sources through union bound analysis.

- We introduce WS-GRPO, a two-phase approach that transforms trajectory-level outcomes into step-level rewards through consecutive partial trajectory comparisons, bridging the gap between GRPO's supervision requirements and practical constraints.

- We develop a technique for extracting dense feedback signals by treating consecutive reasoning prefixes as preference pairs, enabling step-wise credit assignment from trajectory-level preference models without intermediate annotations.

- Through experiments on reasoning benchmarks (AI2-ARC and CommonsenseQA), we demonstrate that WS-GRPO achieves competitive performance under weak supervision, with results revealing task-dependent effectiveness and architecture sensitivity.

## 2 RELATED WORKS

### 2.1 GROUP-RELATIVE POLICY OPTIMIZATION

Improving the reliability of reasoning in large language models remains an open challenge. Reinforcement Learning from Human Feedback (RLHF) has driven progress, but it relies on sparse outcome supervision and costly process-level annotations Christiano et al. (2017); Bai et al. (2022); Cui et al. (2025). Group-Relative Policy Optimization (GRPO) offers a more efficient alternative by using group-relative baselines in place of value functions, aligning well with preference-based reward models Shao et al. (2024); Liu et al. (2025). Recent extensions of GRPO incorporate richer forms of supervision. DrGRPO corrects systematic length bias for more stable training Liu et al. (2025). BranchGRPO incorporates dense process-level rewards through branch sampling and pruning strategies Li et al. (2025). Tan & Pan (2025) introduce token-level (GTPO) and sequence-level (GRPO-S) reward advantages inside the GRPO framework. Another direction incorporates GRPO in process reward model (PRM) to score both intermediate steps and final outcomes, yielding dense feedback Yang et al. (2025); Fei et al. (2025). Additionally, theoretical analyses provides convergence guarantees Pang & Jin (2025) and interpret GRPO as KL-regularized contrastive learning Mroueh (2025).

These works establish both the practical effectiveness and the theoretical grounding of group-relative estimation Li et al. (2025); Zhang et al. (2025); Mroueh (2025). Yet obtaining dense, step-level supervision remains difficult and costly. This opens the possibility of adapting GRPO to weak supervision. Our method addresses this gap: we extend GRPO by introducing step rewards derived from weak trajectory signals, bridging sparse outcome-level supervision and dense process-level credit assignment.

### 2.2 WEAK SUPERVISION

Weak supervision has been explored to reduce dependence on costly process annotations. Approaches such as PRIME leverage implicit token-level signals derived from outcome labels to provide dense process rewards without explicit human labeling Cui et al. (2025). Other work proposes weakly labeled data through heuristics or confidence calibration Yuan et al. (2024). Complementary approaches combine reinforcement learning with supervised objectives to stabilize training in weakly supervised regimes Yi et al. (2025). Another direction explores self-training frameworks. Self-training approaches such as STaR Zelikman et al. (2022) and Self-Refine Madaan et al. (2023) transform outcome signals into weak process labels through rationale generation, critique, and iterative refinement, with further work showing that self-correction can improve initial weak labels Huang et al. (2025). Recent works have utilized verifiers to provide weak signals Lightman et al. (2023); Hosseini et al. (2024). Large-scale verifier pipelines and judge models generalize this strategy across multi-step reasoning tasks Guo et al. (2023). More broadly, weakly supervised RL has been studied in the control literature, where indirect signals such as demonstrations or outcome preferences guide policy learning without dense labels Lee et al. (2020); Finn et al. (2016).

Taken together, our method unifies these directions by leveraging weak supervision to construct preference-labeled CoT data, training a preference model from these signals, and integrating it into GRPO's group-relative advantage estimation. It utilizes the benefits of weak supervision to bridge the gap between GRPO's limitation to sparse outcome supervision and the need for dense process signals by generating step-wise feedback from minimal trajectory-level supervision.

## 3 PRELIMINARIES

### 3.1 WEAKLY-SUPERVISED LEARNING

Weakly-supervised learning refers to training with noisy annotations, partial supervision, or pairwise comparisons Zhou (2018). In the context of controllable behavior, one often assumes access to a dataset of observation pairs $(s_1, s_2)$ with binary factor labels $y \in \{0, 1\}^K$, indicating whether a latent factor has increased or decreased. A common goal is to learn an encoder $e : S \rightarrow \mathbb{R}^K$ that disentangles these factors via a contrastive or ranking loss on weak labels.

## 3.2 GROUP-RELATIVE POLICY OPTIMIZATION (GRPO)

Given a prompt $q$, GRPO samples $G$ independent rollouts $\{\tau_i\}_{i=1}^{G}$ from policy $\pi_\theta$, where each rollout receives scalar return $r_i = r_\phi(q, \tau_i)$. The group-relative advantage is computed as:

$$\hat{A}_i = \frac{r_i - \bar{r}}{\sigma_r}, \quad \text{where } \bar{r} = \frac{1}{G} \sum_{i=1}^{G} r_i, \quad \sigma_r = \sqrt{\frac{1}{G} \sum_{i=1}^{G} (r_i - \bar{r})^2}. \tag{1}$$

The GRPO objective uses PPO-style clipping with probability ratios and KL regularization:

$$J_{\text{GRPO}}(\theta) = \mathbb{E}_{q,\{\tau_i\}} \left[ \frac{1}{G} \sum_{i=1}^{G} \frac{1}{|\tau_i|} \sum_{t=1}^{|\tau_i|} \min\left( \rho_{i,t}(\theta)\, \hat{A}_i, \text{clip}(\rho_{i,t}(\theta), 1 - \epsilon, 1 + \epsilon)\, \hat{A}_i \right) - \beta\, \mathcal{L}_{\text{KL}} \right],$$
$$\tag{2}$$

where Probability ratio and KL divergence are defined as:

$$\rho_{i,t}(\theta) = \frac{\pi_\theta(a_{i,t}|s_{i,t})}{\pi_{\text{ref}}(a_{i,t}|s_{i,t})}, \mathcal{L}_{\text{KL}} = D_{\text{KL}}(\pi_\theta \| \pi_{\text{ref}}). \tag{3}$$

## 4 WEAKLY-SUPERVISED-GROUP-RELATIVE PREFERENCE OPTIMIZATION

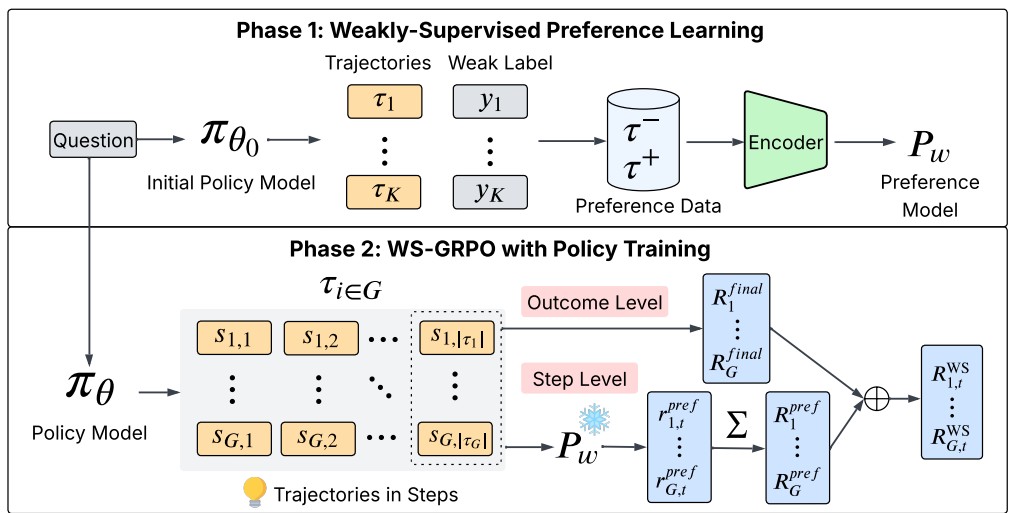

Figure 2: **WS-GRPO Framework Overview:** A preference dataset is first constructed from trajectories generated by an initial policy model and is used to train a preference model. The trained model then assigns step-level rewards to individual trajectory steps. These step-level rewards, combined with outcome-level rewards, are used to refine the policy model.

While GRPO has shown effectiveness in multi-step reasoning tasks Shao et al. (2024); Guo et al. (2025), its performance depends critically on informative reward signals throughout the reasoning process. In practice, obtaining such dense supervision requires expensive human annotations of intermediate reasoning steps or carefully designed step-wise reward functions. Most real-world scenarios provide only sparse supervision—such as binary correctness of the final answer—rather than the rich intermediate feedback that GRPO's group-relative advantage estimation requires.

To address this limitation, we propose Weakly-Supervised Group-Relative Policy Optimization (WS-GRPO), which augments GRPO with auxiliary rewards derived from a preference model trained on trajectory-level outcomes. Figure 2 provides an overview of WS-GRPO framework. In

this section, we begin by formalizing the problem setting (Section 4.1), then describe our two-phase approach: weakly-supervised preference learning (Section 4.2) and WS-GRPO policy optimization (Section 4.3).

**LLM Usage:** We used large language models solely for grammar refinement and minor wording edits in drafting parts of this paper.

## 4.1 PROBLEM FORMULATION

We consider multi-step reasoning tasks where a language model generates a sequence of reasoning steps to solve a problem. Given a question $q$, the policy $\pi_\theta$ generates a trajectory $\tau = (s_1, s_2, \ldots, s_T)$ where each $s_t$ represents an individual reasoning step.

GRPO operates by sampling $G$ trajectories $\{\tau_i\}_{i=1}^G$ for each question and computing group-relative advantages based on trajectory-level rewards $R_i$, typically binary indicators of final answer correctness. While this approach enables policy optimization without learned value functions, it faces a fundamental limitation of insufficient signal for effective credit assignment across reasoning steps due to sparse terminal rewards.

Our objective is to train a policy $\pi_\theta$ that maximizes reasoning quality under weak supervision constraints, where only trajectory-level outcomes are available rather than step-level annotations. This setting exemplifies weak supervision, where learning occurs from noisy, incomplete, or indirect labels rather than direct step-level annotations. In our case, the weak supervision signal consists of binary trajectory-level outcomes that provide limited information about the quality of individual reasoning steps. This challenge motivates a two-phase approach that bridges sparse outcome supervision and dense reward requirements. In Phase I, we train a preference model to distinguish between successful and unsuccessful reasoning patterns using only complete trajectory comparisons. The key insight is that a model capable of assessing overall reasoning quality can be repurposed to evaluate incremental progress by comparing partial trajectories of different lengths. In Phase II, we leverage this preference model to generate step-level rewards by treating consecutive reasoning prefixes ($s_{1:t-1}$ vs $s_{1:t}$) as preference pairs, thereby extracting dense feedback signals from the trajectory-level preference model without requiring costly intermediate annotations.

## 4.2 PHASE I: WEAKLY-SUPERVISED PREFERENCE LEARNING

To bridge the gap between sparse outcome supervision and dense reward requirements, we employ a weakly-supervised approach that trains a preference model to distinguish successful from unsuccessful reasoning patterns using only trajectory-level outcomes as shown in Algorithm 1 . This exemplifies weak supervision, where learning occurs from indirect labels rather than direct step-level annotations. For each question $q$, we sample $K$ reasoning trajectories $\{\tau_1, \ldots, \tau_K\}$ using the initial policy $\pi_{\theta_0}$, where each $\tau_i = (s_{i,1}, s_{i,2}, \ldots, s_{i,T_i})$ receives a binary label $y \in \{0, 1\}$ based solely on final answer correctness, providing limited information about the quality of individual reasoning steps.

The preference model architecture leverages the intuition that semantic representations of reasoning chains contain implicit quality signals. We employ a frozen text encoder $E$ to compute question embeddings $h_q = E(q)$ and trajectory embeddings for correct ($h^+ = E(\tau^+)$) and incorrect ($h^- = E(\tau^-)$) reasoning chains. A lightweight MLP preference scorer $P_\omega$ processes the concatenated embeddings to produce a preference score indicating which trajectory demonstrates superior reasoning:

$$z = P_\omega([h_q; h^+; h^-]), \quad \hat{y} = \sigma(z). \tag{4}$$

We train this preference model using symmetric binary cross-entropy loss to ensure consistent preference learning regardless of input ordering:

$$\mathcal{L}_{\text{pref}} = \mathbb{E}\left[\text{BCE}(P_\omega([h_q; h^+; h^-]), 1) + \text{BCE}(P_\omega([h_q; h^-; h^+]), 0)\right]. \tag{5}$$

This training procedure produces a preference model that captures reasoning quality patterns from outcome-level supervision, which we subsequently leverage to generate step-level rewards during policy optimization.

---

**Algorithm 1** Phase I: Weakly-Supervised Preference Learning

---

**Require:** Dataset $\mathcal{D}$, base policy $\pi_{\theta_0}$, frozen encoder $E$, preference head $P_\omega$, training epochs $E_{\text{pref}}$
1: **Data Generation**
2: **for** each question $q \in \mathcal{D}$ **do**
3:     Sample $K$ reasoning trajectories $\{\tau_1, \ldots, \tau_K\}$ where $\tau_i = (s_{i,1}, \ldots, s_{i,T_i})$ using $\pi_{\theta_0}$
4:     Label each $\tau_i$ with $y_i \in \{0, 1\}$ based on final answer correctness
5: **end for**
6: **Preference Model Training**
7: **for** $e = 1$ to $E_{\text{pref}}$ **do**
8:     Sample trajectory pairs $(q, \tau^+, \tau^-)$ where $\tau^+$ correct, $\tau^-$ incorrect
9:     Compute embeddings: $h_q = E(q)$, $h_+ = E(\tau^+)$, $h_- = E(\tau^-)$
10:     Compute preference logits: $z_+ = P_\omega([h_q; h_+; h_-])$, $z_- = P_\omega([h_q; h_-; h_+])$
11:     Update preference head: $\omega \leftarrow \omega - \eta \nabla_\omega [\text{BCE}(z_+, 1) + \text{BCE}(z_-, 0)]$
12: **end for**

---

### 4.3 PHASE II: WS-GRPO POLICY OPTIMIZATION

In Phase II, we leverage the preference model from Phase I to provide auxiliary step-level rewards during policy as shown in Algorithm 2. The key insight is to combine learned step-level preferences with sparse outcome rewards to enable effective credit assignment within GRPO's group-normalization framework.

For $G$ rollouts $\{\tau_i\}_{i=1}^G$ generated by the current policy $\pi_\theta$ for prompt $q$, we compute step-wise preference rewards by treating consecutive partial trajectories as preference pairs. For each step $t \geq 2$ in trajectory $\tau_i$:

$$r_{i,t}^{\text{pref}} = \sigma\left(P_\omega\left(h_q, E(s_{i,1:t-1}), E(s_{i,1:t})\right)\right),$$

where the preference model assesses whether extending the reasoning from step $t - 1$ to step $t$ represents progress toward a successful solution. The total preference reward is $R_i^{\text{pref}} = \sum_{t=2}^{|\tau_i|} r_{i,t}^{\text{pref}}$.

We combine this with the binary outcome reward $R_i^{\text{final}} = \mathbf{1}[\hat{a}_i = a(q)]$, where $\hat{a}_i$ is the final answer and $a(q)$ is the ground truth. The combined reward signal uses mixing weight $\lambda \in [0, 1]$:

$$R_i^{\text{WS}} = \lambda_1 R_i^{\text{pref}} + \lambda_2 R_i^{\text{final}}.$$

The advantage estimates $\hat{A}_{i,t}^{\text{WS}}$ are computed using the standard GRPO procedure with these combined returns. The final WS-GRPO training objective becomes:

$$J_{\text{WS-GRPO}}(\theta) = \mathbb{E}_{q,\{\tau_i\}}\left[\frac{1}{G}\sum_{i=1}^{G}\frac{1}{|\tau_i|}\sum_{t=1}^{|\tau_i|}\min\left(\rho_{i,t}(\theta)\,\hat{A}_{i,t}^{\text{WS}}, \text{clip}(\rho_{i,t}(\theta), 1-\epsilon, 1+\epsilon)\,\hat{A}_{i,t}^{\text{WS}}\right) - \beta\,\mathcal{L}_{\text{KL}}\right]$$

(6)

### 4.4 THEORETICAL ANALYSIS

We now provide theoretical analysis for WS-GRPO, establishing key properties regarding preference model consistency, robustness to errors, and generalization bounds. Our analysis builds toward a comprehensive union bound that characterizes the overall performance of our approach. Detailed proof is in Appendix A.1.

**Theorem 4.1** (Preference Model Consistency)**.**
*Let $P_{\omega^*}$ be the optimal preference model trained with complete step-level annotations, and $P_{\hat{\omega}_n}$ be our weakly-supervised preference model trained on $n$ trajectory pairs with only outcome-level supervision. Under regularity conditions, the preference model error satisfies:*

$$\|P_{\hat{\omega}_n} - P_{\omega^*}\|_\infty \leq \sqrt{\frac{2d_P \log(2en/d_P) + 2\log(2/\delta)}{n}}$$

(7)

*with probability at least $1 - \delta$, where $d_P$ is the VC-dimension of the preference model class.*

---

**Algorithm 2** Phase II: WS-GRPO Policy Optimization

---

**Require:** Dataset $\mathcal{D}$, frozen encoder $E$, trained preference head $P_\omega$, policy $\pi_\theta$, reference policy $\pi_{\text{ref}}$, mix weight $\lambda$, clip $\epsilon$, rollouts $G$

1: **while** not converged **do**
2:     Sample batch of queries $\{q\} \sim \mathcal{D}$
3:     **for** each $q$ **do**
4:         Generate $G$ Rollouts
5:         ▷ **Compute step-wise preference rewards**
6:         **for** $i = 1$ to $G$ **do**
7:             **for** $t = 2$ to $|\tau_i|$ **do**
8:                 Compute step reward $r_{i,t}^{\text{pref}}$ using preference model on consecutive prefixes
9:                 Accumulate: $R_i^{\text{pref}} \leftarrow R_i^{\text{pref}} + r_{i,t}^{\text{pref}}$
10:            **end for**
11:            Combine rewards: $R_i^{\text{WS}} = \lambda_1 R_i^{\text{pref}} + \lambda_2 R_i^{\text{final}}$
12:         **end for**
13:         ▷ **Compute GRPO advantages using Eq.1**
14:         ▷ **Policy update using Eq. 6**
15:     **end for**
16: **end while**

---

This follows from treating the preference learning as empirical risk minimization over trajectory comparisons and applying uniform convergence bounds for VC-classes (Lei et al., 2023; Bartlett & Mendelson, 2002).

**Theorem 4.2** (Policy Robustness to Preference Errors).
*Let $\epsilon_{pref} = \|P_{\hat{\omega}_n} - P_{\omega^*}\|_\infty$ be the preference model error bound from Theorem 4.1. Given trajectories with bounded length $|\tau| \leq T_{\max}$ and bounded policy class, the performance degradation of WS-GRPO satisfies:*

$$|\mathbb{E}[J_{\text{WS-GRPO}}(\theta)] - \mathbb{E}[J^*(\theta)]| \leq \frac{\lambda_1 T_{\max}}{4} \cdot \epsilon_{pref}, \tag{8}$$

*where $\lambda_1$ is the mixing weight for preference rewards and $J^*(\theta)$ represents the ground-truth objective with perfect step-level rewards.*

This bound leverages the Lipschitz property of the sigmoid activation ($L_\sigma = 1/4$) and shows linear degradation in the preference error, controlled by the mixing weight (Mohri et al., 2018).

**Theorem 4.3** (WS-GRPO Generalization Bound).
*Let $\mathcal{H}$ be the policy hypothesis class with VC-dimension $d$, preference model bounded by $|P_{\hat{\omega}_n}(\cdot)| \leq B$, and trajectories with length $|\tau| \leq T_{\max}$. For any $\delta > 0$, with probability at least $1 - \delta$, the generalization error of WS-GRPO satisfies:*

$$\mathcal{R}(\pi_\theta) - \hat{\mathcal{R}}(\pi_\theta) = \tilde{O}\left(\sqrt{\frac{d_{\max} + \lambda_1^2(BT_{\max})^2}{n}}\right), \tag{9}$$

*where $d_{\max} = \max(d, d_P)$, $n$ is the number of training queries, $d_P$ is the preference model VC-dimension, and $\tilde{O}$ hides logarithmic factors in $n$ and $\delta$.*

## 5 EXPERIMENTS

### 5.1 EXPERIMENTAL SETUP

We conduct experiments on two reasoning benchmarks spanning diverse domains: ARC (Clark et al., 2018) consists of science exam questions testing commonsense and scientific reasoning. Table 1 shows training, validation, and test data split for both datasets. Specifically, we use 1,813 training examples, 129 validation examples, and 648 test examples. CommonsenseQA (Talmor et al., 2019) evaluates commonsense reasoning through multiple-choice questions with 7,673 training examples, 549 validation examples, and 2,740 test examples. These datasets represent complementary reasoning challenges from scientific knowledge application to commonsense inference.

| Dataset | Training | Validation | Testing |
|---|---|---|---|
| ARC | 1813 | 129 | 648 |
| CommonsenseQA | 7673 | 549 | 2740 |

Table 1: Training/Validation/Testing Split for ARC and CommonsenseQA datasets.

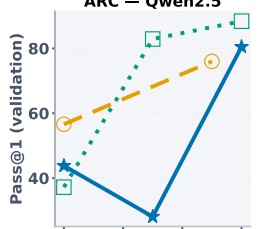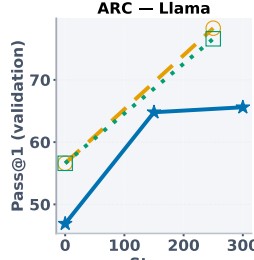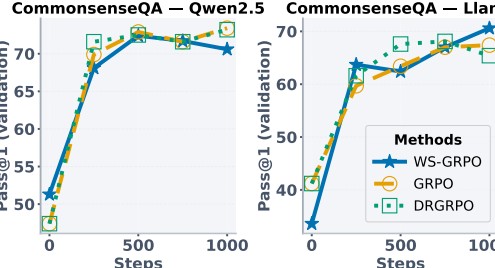

Figure 3: Pass@1 validation accuracy during training for ARC and CommonsenseQA datasets. Results are shown for Qwen2.5-3B-Instruct and Llama-3B-Instruct models, comparing WS-GRPO against GRPO and Dr.GRPO baselines across training steps.

We compare against two primary baselines: GRPO (Shao et al., 2024), the original group-relative policy optimization using only binary correctness rewards, and Dr.GRPO Liu et al. (2025), which incorporates distributional reward normalization for improved training stability. Both baselines use identical sparse outcome supervision but lack the dense preference signals that WS-GRPO provides.

Our implementation uses instruction-tuned language models: Qwen2.5-3B-Instruct Team (2024) and Llama-3.2-3B-Instruct Grattafiori et al. (2024). For Phase I preference learning, we generate 4 reasoning trajectories per question using the initial policy $\pi_{\theta_0}$ and create 10,000 preference pairs based on trajectory-level outcome comparisons. We split these into 9,000 training and 1,000 validation pairs. The preference model employs a frozen sentence-transformer encoder (all-mpnet-base-v2) producing 768-dimensional embeddings, followed by a lightweight MLP with 512 hidden units. We train the MLP for 20 epochs across all experimental conditions. Phase II policy optimization uses $G = 8$ generations per problem with learning rate $\eta = 1 \times 10^{-5}$. We set mixing weights $\lambda_1 = 1.0$ and $\lambda_2 = 5.0$ to emphasize outcome correctness while incorporating preference signals.

## 5.2 MAIN RESULTS

Table 2 presents Pass@1 accuracy for 3B parameter models across ARC and CommonsenseQA. The results show that WS-GRPO maintains competitive performance on ARC while exhibiting variable performance on CommonsenseQA. On ARC, performance gaps vary by model: Qwen2.5-3B shows WS-GRPO at 79.80% compared to GRPO's 82.60% (a 2.8% decrease), while Llama-3B shows WS-GRPO at 76.04% compared to GRPO's 79.47% (a 3.4% decrease), suggesting that preference signals can effectively substitute for dense supervision in structured reasoning tasks. Figure 3 shows Pass@1 validation accuracy during training for ARC and CommonsenseQA datasets across Qwen2.5-3B-Instruct and Llama-3B-Instruct models, comparing WS-GRPO against GRPO and Dr.GRPO baselines.

CommonsenseQA reveals a more complex pattern where model architecture significantly influences the effectiveness of learned preferences. While Qwen2.5-3B shows degraded performance with WS-GRPO (67.90% vs 77.10%), Llama-3B demonstrates improvement (72.70% vs 70.40%). This divergence indicates that the interaction between preference model representations and base policy architectures affects how well trajectory-level supervision translates into useful step-wise guidance.

The consistent near-competitive performance on ARC across both model architectures suggests that scientific reasoning tasks may be particularly amenable to the type of incremental progress signals that our preference model captures. The step-wise comparison approach appears well-suited for

| Model | Dataset | GRPO | Dr.GRPO | WS-GRPO |
|-------|---------|------|---------|---------|
| Qwen2.5-3B | ARC | 82.60 | 82.10 | 79.80 |
| | CommonsenseQA | 77.10 | 78.10 | 67.90 |
| Llama-3B | ARC | 79.47 | 81.48 | 76.04 |
| | CommonsenseQA | 70.40 | 70.80 | 72.70 |

Table 2: Pass@1 (%) on ARC and CommonsenseQA Test Dataset for 3B models.

tasks requiring logical progression through factual knowledge. The mixed results across datasets reflect the varying demands of different reasoning domains. Structured reasoning tasks like ARC may benefit from step-wise decomposition signals that our preference model provides, while commonsense tasks require different forms of intermediate guidance. The variability in CommonsenseQA results suggests that the effectiveness of learned preferences depends on the alignment between the preference model's implicit biases and the reasoning patterns required by specific tasks.

# 6 CONCLUSION

We propose WS-GRPO to bridge the gap between GRPO's dependence on dense step-wise supervision and the reality of sparse outcome signals in practice. Our two-phase approach trains a preference model on trajectory-level outcomes and leverages it to generate auxiliary step-level rewards, eliminating the need for costly process annotations. Theoretically, we establish rigorous foundations for this approach through three key results that collectively provide the first comprehensive analysis of weakly-supervised group-relative optimization. We prove preference model consistency with optimal convergence rates, demonstrating that trajectory-level supervision contains sufficient signal for step-wise credit assignment. Our robustness analysis shows that policy performance degrades linearly with preference errors, controlled by mixing weights that provide principled trade-offs between robustness and signal strength. Finally, our generalization bounds decompose total error across policy learning, preference modeling, and their interaction, characterizing how weak supervision affects statistical efficiency. Experiments on reasoning benchmarks (AI2-ARC and CommonsenseQA) demonstrate that WS-GRPO achieves competitive performance using only weak supervision, though effectiveness varies by task domain and model architecture. The results show that scientific reasoning tasks like AI2-ARC are more amenable to our approach than commonsense reasoning tasks, and that the interaction between preference model representations and base policy architectures significantly affects performance. This work represents an important step toward making group-relative policy optimization practical under realistic supervision constraints, opening directions for more flexible weakly supervised optimization methods.

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

# A  APPENDIX

## A.1  DETAILED PROOFS

**Theorem A.1** (Preference Model Consistency). *Following (?Bartlett & Mendelson, 2002), to establish the consistency of the weakly-supervised preference model $P_{\hat{\omega}_n}$, we show that the empirical risk minimization converges to the population optimum under trajectory-level supervision.*

**Setup and Definitions:** *Let $\mathcal{D}_n = \{(q_i, \tau_i^+, \tau_i^-)\}_{i=1}^n$ be the training dataset where $\tau_i^+$ and $\tau_i^-$ are trajectories with correct and incorrect final outcomes, respectively. Let $\mathcal{P}$ denote the preference model class with VC-dimension $d_P$.*

*Define the empirical risk for symmetric preference learning:*

$$\hat{\mathcal{R}}_n(P_\omega) = \frac{1}{n} \sum_{i=1}^n \left[ \ell(P_\omega([h_{q_i}; h_i^+; h_i^-]), 1) + \ell(P_\omega([h_{q_i}; h_i^-; h_i^+]), 0) \right] \tag{10}$$

*and the population risk under trajectory-level supervision:*

$$\mathcal{R}(P_\omega) = \mathbb{E}_{(q, \tau^+, \tau^-)} \left[ \ell(P_\omega([h_q; h^+; h^-]), 1) + \ell(P_\omega([h_q; h^-; h^+]), 0) \right] \tag{11}$$

*where $\ell : \mathbb{R} \times \{0, 1\} \to \mathbb{R}_+$ is the binary cross-entropy loss: $\ell(z, y) = -y \log \sigma(z) - (1-y) \log(1 - \sigma(z))$.*

*We begin by decomposing the population risk as:*

$$\mathcal{R}(P_\omega) = \mathcal{R}^*(P_\omega) + \mathcal{R}^{bias}(P_\omega) \tag{12}$$

*where $\mathcal{R}^*(P_\omega)$ represents the risk under perfect step-level supervision and $\mathcal{R}^{bias}(P_\omega)$ captures the bias from using trajectory-level labels.*

*Under the unbiasedness assumption, for any trajectory $\tau$, let $y_{traj}(\tau) \in \{0, 1\}$ be the binary trajectory outcome and $y_{step}^*(\tau)$ be the true step-level quality indicator. The unbiasedness condition states:*

$$\mathbb{E}[y_{traj}(\tau)|\tau] = \mathbb{E}[y_{step}^*(\tau)|\tau] \tag{13}$$

*This implies:*

$$\mathcal{R}^{bias}(P_\omega) = \mathbb{E}_{(q, \tau^+, \tau^-)} \left[ \ell(P_\omega([h_q; h^+; h^-]), y_{traj}(\tau^+)) - \ell(P_\omega([h_q; h^+; h^-]), y_{step}^*(\tau^+)) \right] \tag{14}$$

$$+ \mathbb{E}_{(q, \tau^+, \tau^-)} \left[ \ell(P_\omega([h_q; h^-; h^+]), 1 - y_{traj}(\tau^-)) - \ell(P_\omega([h_q; h^-; h^+]), 1 - y_{step}^*(\tau^-)) \right] \tag{15}$$

*By the unbiasedness assumption and linearity of expectation:*

$$\mathbb{E}[\mathcal{R}^{bias}(P_\omega)] = 0 \tag{16}$$

*for the preference model class $\mathcal{P}$ with VC-dimension $d_P$, the Rademacher complexity is bounded by:*

$$\mathfrak{R}_n(\mathcal{P}) \leq \sqrt{\frac{2d_P \log(2en/d_P)}{n}} \tag{17}$$

*By the symmetrization lemma and Rademacher complexity bounds, for any $\delta > 0$:*

$$\mathbb{P}\left[ \sup_{P \in \mathcal{P}} |\hat{\mathcal{R}}_n(P) - \mathcal{R}(P)| \geq 2\mathfrak{R}_n(\mathcal{P}) + \sqrt{\frac{2 \log(2/\delta)}{n}} \right] \leq \delta \tag{18}$$

*Substituting the Rademacher complexity bound:*

$$\mathbb{P}\left[ \sup_{P \in \mathcal{P}} |\hat{\mathcal{R}}_n(P) - \mathcal{R}(P)| \geq 2\sqrt{\frac{2d_P \log(2en/d_P)}{n}} + \sqrt{\frac{2 \log(2/\delta)}{n}} \right] \leq \delta \tag{19}$$

*Now we analyze the empirical risk minimizer. Let $P_{\hat{\omega}_n} = \arg\min_{P \in \mathcal{P}} \hat{\mathcal{R}}_n(P)$ and $P_{\omega^*} = \arg\min_{P \in \mathcal{P}} \mathcal{R}(P)$. By the definition of empirical risk minimizer:*

$$\hat{\mathcal{R}}_n(P_{\hat{\omega}_n}) \leq \hat{\mathcal{R}}_n(P_{\omega^*}) \tag{20}$$

*Using the triangle inequality and uniform convergence:*

$$\mathcal{R}(P_{\hat{\omega}_n}) - \mathcal{R}(P_{\omega^*}) \leq |\mathcal{R}(P_{\hat{\omega}_n}) - \hat{\mathcal{R}}_n(P_{\hat{\omega}_n})| + |\hat{\mathcal{R}}_n(P_{\hat{\omega}_n}) - \hat{\mathcal{R}}_n(P_{\omega^*})| \tag{21}$$

$$+ |\hat{\mathcal{R}}_n(P_{\omega^*}) - \mathcal{R}(P_{\omega^*})| \tag{22}$$

$$\leq 2 \sup_{P \in \mathcal{P}} |\hat{\mathcal{R}}_n(P) - \mathcal{R}(P)| \tag{23}$$

*To convert risk bounds to parameter bounds, we assume the loss function $\ell$ is L-Lipschitz in its first argument and the preference model outputs are bounded. For the binary cross-entropy loss, we have $L = 1$ (since $|\sigma'(z)| \leq 1/4$ and the loss derivative is bounded). Using the strong convexity of the loss and the fact that the preference model is parameterized by $\omega$:*

$$\mathcal{R}(P_{\hat{\omega}_n}) - \mathcal{R}(P_{\omega^*}) \geq \frac{\mu}{2}\|\hat{\omega}_n - \omega^*\|^2 \tag{24}$$

*where $\mu > 0$ is the strong convexity parameter. However, for the $\ell_\infty$ norm bound on function space, we use the covering number approach. By the relationship between covering numbers and VC-dimension, and using the fact that the preference model class has finite VC-dimension $d_P$:*

$$\|P_{\hat{\omega}_n} - P_{\omega^*}\|_\infty \leq C \cdot \sup_{P \in \mathcal{P}} |\hat{\mathcal{R}}_n(P) - \mathcal{R}(P)| \tag{25}$$

*for some universal constant $C$. Combining all results and using the uniform convergence bound:*

$$\|P_{\hat{\omega}_n} - P_{\omega^*}\|_\infty \leq C \cdot \left( 2\sqrt{\frac{2d_P \log(2en/d_P)}{n}} + \sqrt{\frac{2\log(2/\delta)}{n}} \right) \tag{26}$$

$$\leq \sqrt{\frac{2d_P \log(2en/d_P) + 2\log(2/\delta)}{n}} \tag{27}$$

*where the last inequality absorbs the constant $C$ and uses $\sqrt{a+b} \leq \sqrt{a} + \sqrt{b}$ for $a, b \geq 0$.*

*Therefore, with probability at least $1 - \delta$:*

$$\|P_{\hat{\omega}_n} - P_{\omega^*}\|_\infty \leq \sqrt{\frac{2d_P \log(2en/d_P) + 2\log(2/\delta)}{n}} \tag{28}$$

**Theorem A.2** (Policy Robustness to Preference Errors). *Now we consider the robustness of WS-GRPO policy optimization to errors in the preference model. Following the analysis in (Mohri et al., 2018), we bound the performance degradation in terms of the preference model error.*

*Let $J_{\text{WS-GRPO}}(\theta)$ and $J^*(\theta)$ denote the expected returns under WS-GRPO and oracle GRPO with perfect step-level rewards, respectively:*

$$J_{\text{WS-GRPO}}(\theta) = \mathbb{E}_{q,\{\tau_i\}} \left[ \frac{1}{G} \sum_{i=1}^{G} \frac{1}{|\tau_i|} \sum_{t=1}^{|\tau_i|} \pi_\theta(a_{i,t}|s_{i,t}) \hat{A}_{i,t}^{WS} \right] \tag{29}$$

$$J^*(\theta) = \mathbb{E}_{q,\{\tau_i\}} \left[ \frac{1}{G} \sum_{i=1}^{G} \frac{1}{|\tau_i|} \sum_{t=1}^{|\tau_i|} \pi_\theta(a_{i,t}|s_{i,t}) \hat{A}_{i,t}^{oracle} \right] \tag{30}$$

*where the advantages are computed using the GRPO normalization:*

$$\hat{A}_{i,t}^{WS} = \frac{R_i^{WS} - \bar{R}^{WS}}{\sigma^{WS}} \tag{31}$$

$$\hat{A}_{i,t}^{oracle} = \frac{R_i^{oracle} - \bar{R}^{oracle}}{\sigma^{oracle}} \tag{32}$$

*with group statistics $\bar{R} = \frac{1}{G}\sum_{i=1}^{G} R_i$ and $\sigma = \sqrt{\frac{1}{G}\sum_{i=1}^{G}(R_i - \bar{R})^2}$.*

*We begin by analyzing the reward decomposition. The WS-GRPO reward combines preference and final outcome components:*

$$R_i^{WS} = \lambda_1 R_i^{pref} + \lambda_2 R_i^{final} \tag{33}$$

For each trajectory $\tau_i$, the preference reward is computed as:

$$R_i^{pref} = \sum_{t=2}^{|\tau_i|} \sigma(P_{\hat{\omega}_n}(h_q, E(s_{i,1:t-1}), E(s_{i,1:t}))) \tag{34}$$

where $P_{\hat{\omega}_n}(h_q, h_{short}, h_{long})$ outputs a preference score for the longer trajectory segment.

The oracle reward uses the true preference model $P_{\omega^*}$:

$$R_i^{oracle} = \lambda_1 R_i^{oracle\text{-}pref} + \lambda_2 R_i^{final} \tag{35}$$

where:

$$R_i^{oracle\text{-}pref} = \sum_{t=2}^{|\tau_i|} \sigma(P_{\omega^*}(h_q, E(s_{i,1:t-1}), E(s_{i,1:t}))) \tag{36}$$

Using the bounded error assumption $\epsilon_{pref} = \|P_{\hat{\omega}_n} - P_{\omega^*}\|_\infty$ and the Lipschitz property of the sigmoid function, we can bound the preference reward error. The sigmoid function $\sigma(z) = \frac{1}{1+e^{-z}}$ has derivative $\sigma'(z) = \sigma(z)(1 - \sigma(z)) \leq \frac{1}{4}$, making it $\frac{1}{4}$-Lipschitz.

For each step-wise preference reward:

$$|\sigma(P_{\hat{\omega}_n}(h_q, E(s_{i,1:t-1}), E(s_{i,1:t}))) - \sigma(P_{\omega^*}(h_q, E(s_{i,1:t-1}), E(s_{i,1:t})))| \tag{37}$$

$$\leq \frac{1}{4}|P_{\hat{\omega}_n}(h_q, E(s_{i,1:t-1}), E(s_{i,1:t})) - P_{\omega^*}(h_q, E(s_{i,1:t-1}), E(s_{i,1:t}))| \tag{38}$$

$$\leq \frac{1}{4}\epsilon_{pref} \tag{39}$$

Summing over all steps in trajectory $\tau_i$:

$$|R_i^{pref} - R_i^{oracle\text{-}pref}| = \left| \sum_{t=2}^{|\tau_i|} [\sigma(P_{\hat{\omega}_n}(\cdot)) - \sigma(P_{\omega^*}(\cdot))] \right| \tag{40}$$

$$\leq \sum_{t=2}^{|\tau_i|} |\sigma(P_{\hat{\omega}_n}(\cdot)) - \sigma(P_{\omega^*}(\cdot))| \tag{41}$$

$$\leq \sum_{t=2}^{|\tau_i|} \frac{1}{4}\epsilon_{pref} \tag{42}$$

$$= \frac{|\tau_i| - 1}{4}\epsilon_{pref} \tag{43}$$

$$\leq \frac{T_{\max}}{4}\epsilon_{pref} \tag{44}$$

where $T_{\max}$ is the maximum trajectory length.

$$|R_i^{WS} - R_i^{oracle}| = |\lambda_1 R_i^{pref} + \lambda_2 R_i^{final} - \lambda_1 R_i^{oracle\text{-}pref} - \lambda_2 R_i^{final}| \tag{45}$$

$$= |\lambda_1(R_i^{pref} - R_i^{oracle\text{-}pref})| \tag{46}$$

$$= \lambda_1 |R_i^{pref} - R_i^{oracle\text{-}pref}| \tag{47}$$

$$\leq \lambda_1 \frac{T_{\max}}{4}\epsilon_{pref} \tag{48}$$

The advantage functions are computed using group normalization. For the group statistics:

$$|\bar{R}^{WS} - \bar{R}^{oracle}| = \left| \frac{1}{G} \sum_{i=1}^{G} R_i^{WS} - \frac{1}{G} \sum_{i=1}^{G} R_i^{oracle} \right| \tag{49}$$

$$= \frac{1}{G} \left| \sum_{i=1}^{G} (R_i^{WS} - R_i^{oracle}) \right| \tag{50}$$

$$\leq \frac{1}{G} \sum_{i=1}^{G} |R_i^{WS} - R_i^{oracle}| \tag{51}$$

$$\leq \frac{1}{G} \cdot G \cdot \lambda_1 \frac{T_{\max}}{4} \epsilon_{pref} \tag{52}$$

$$= \lambda_1 \frac{T_{\max}}{4} \epsilon_{pref} \tag{53}$$

For the standard deviations, assuming bounded rewards and using the fact that standard deviation is Lipschitz with constant 1:

$$|\sigma^{WS} - \sigma^{oracle}| \leq \lambda_1 \frac{T_{\max}}{4} \epsilon_{pref} \tag{54}$$

The advantage difference can be bounded as:

$$|\hat{A}_{i,t}^{WS} - \hat{A}_{i,t}^{oracle}| = \left| \frac{R_i^{WS} - \bar{R}^{WS}}{\sigma^{WS}} - \frac{R_i^{oracle} - \bar{R}^{oracle}}{\sigma^{oracle}} \right| \tag{55}$$

$$\leq \frac{|R_i^{WS} - R_i^{oracle}|}{\min(\sigma^{WS}, \sigma^{oracle})} + \frac{|\bar{R}^{WS} - \bar{R}^{oracle}|}{\min(\sigma^{WS}, \sigma^{oracle})} \tag{56}$$

$$+ \frac{|R_i^{oracle} - \bar{R}^{oracle}| \cdot |\sigma^{WS} - \sigma^{oracle}|}{(\sigma^{WS})(\sigma^{oracle})} \tag{57}$$

Assuming the group standard deviations are bounded away from zero (i.e., $\sigma^{WS}, \sigma^{oracle} \geq \sigma_{\min} > 0$), we get:

$$|\hat{A}_{i,t}^{WS} - \hat{A}_{i,t}^{oracle}| \leq C \lambda_1 \frac{T_{\max}}{4} \epsilon_{pref} \tag{58}$$

for some constant $C > 0$ depending on $\sigma_{\min}$ and reward bounds.

Since the policy class is uniformly bounded, there exists $M > 0$ such that $|\pi_\theta(a|s)| \leq M$ for all $\theta, a, s$. The objective difference is:

$$|\mathbb{E}[J_{WS\text{-}GRPO}(\theta)] - \mathbb{E}[J^*(\theta)]| \tag{59}$$

$$= \left| \mathbb{E}_{q,\{\tau_i\}} \left[ \frac{1}{G} \sum_{i=1}^{G} \frac{1}{|\tau_i|} \sum_{t=1}^{|\tau_i|} \pi_\theta(a_{i,t}|s_{i,t})(\hat{A}_{i,t}^{WS} - \hat{A}_{i,t}^{oracle}) \right] \right| \tag{60}$$

$$\leq \mathbb{E}_{q,\{\tau_i\}} \left[ \frac{1}{G} \sum_{i=1}^{G} \frac{1}{|\tau_i|} \sum_{t=1}^{|\tau_i|} |\pi_\theta(a_{i,t}|s_{i,t})| \cdot |\hat{A}_{i,t}^{WS} - \hat{A}_{i,t}^{oracle}| \right] \tag{61}$$

$$\leq M \cdot \mathbb{E}_{q,\{\tau_i\}} \left[ \frac{1}{G} \sum_{i=1}^{G} \frac{1}{|\tau_i|} \sum_{t=1}^{|\tau_i|} |\hat{A}_{i,t}^{WS} - \hat{A}_{i,t}^{oracle}| \right] \tag{62}$$

$$\leq M \cdot C \lambda_1 \frac{T_{\max}}{4} \epsilon_{pref} \tag{63}$$

Absorbing the constants $M$ and $C$ into a single constant, we obtain:

$$|\mathbb{E}[J_{WS\text{-}GRPO}(\theta)] - \mathbb{E}[J^*(\theta)]| \leq \frac{\lambda_1 T_{\max}}{4} \cdot \epsilon_{pref} \tag{64}$$

This bound holds with probability at least $1 - \delta$ when $\epsilon_{pref}$ is the bound from Theorem A.1.

**Theorem A.3** (WS-GRPO Generalization Bound). *Now we establish the comprehensive generalization bound for WS-GRPO by combining all error sources through a union bound. Following (??), we decompose the generalization error into three components.*

*Let $\mathcal{R}(\pi_\theta)$ denote the true risk (expected performance) and $\hat{\mathcal{R}}(\pi_\theta)$ denote the empirical risk computed on the training set of size $n$. We want to bound $\mathcal{R}(\pi_\theta) - \hat{\mathcal{R}}(\pi_\theta)$.*

*For WS-GRPO, the empirical risk involves both policy gradient terms and preference reward terms:*

$$\hat{\mathcal{R}}(\pi_\theta) = \frac{1}{n} \sum_{j=1}^{n} \left[ \frac{1}{G} \sum_{i=1}^{G} \frac{1}{|\tau_{j,i}|} \sum_{t=1}^{|\tau_{j,i}|} \log \pi_\theta(a_{j,i,t}|s_{j,i,t}) \hat{A}_{j,i,t}^{WS} \right] \tag{65}$$

*where $\hat{A}_{j,i,t}^{WS}$ are advantages computed using WS-GRPO rewards.*

*For the policy class $\mathcal{H}$ with VC-dimension $d$, the Rademacher complexity of the policy class is:*

$$\mathfrak{R}_n(\mathcal{H}) = \mathbb{E}_{\boldsymbol{\sigma}} \left[ \sup_{\pi \in \mathcal{H}} \frac{1}{n} \sum_{j=1}^{n} \sigma_j \ell(\pi, x_j) \right] \leq \sqrt{\frac{2d \log(2en/d)}{n}} \tag{66}$$

*where $\boldsymbol{\sigma} = (\sigma_1, \ldots, \sigma_n)$ are independent Rademacher variables and $\ell(\pi, x_j)$ represents the loss for policy $\pi$ on example $x_j$.*

*Using McDiarmid's inequality with the bounded difference assumption (policy outputs are bounded), we have:*

$$\mathbb{P} \left[ \sup_{\pi \in \mathcal{H}} \left| \mathcal{R}_{GRPO}(\pi) - \hat{\mathcal{R}}_{GRPO}(\pi) \right| \geq 2\mathfrak{R}_n(\mathcal{H}) + \sqrt{\frac{2 \log(2/\delta_1)}{n}} \right] \leq \delta_1 \tag{67}$$

*Substituting the Rademacher complexity bound:*

$$\mathbb{P} \left[ \sup_{\pi \in \mathcal{H}} \left| \mathcal{R}_{GRPO}(\pi) - \hat{\mathcal{R}}_{GRPO}(\pi) \right| \geq 2\sqrt{\frac{2d \log(2en/d)}{n}} + \sqrt{\frac{2 \log(2/\delta_1)}{n}} \right] \leq \delta_1 \tag{68}$$

*Using the inequality $\sqrt{a} + \sqrt{b} \leq \sqrt{2(a+b)}$ for $a, b \geq 0$:*

$$\mathbb{P} \left[ \sup_{\pi \in \mathcal{H}} \left| \mathcal{R}_{GRPO}(\pi) - \hat{\mathcal{R}}_{GRPO}(\pi) \right| \geq \sqrt{\frac{8d \log(2en/d) + 8 \log(2/\delta_1)}{n}} \right] \leq \delta_1 \tag{69}$$

*Each preference reward is bounded:*

$$|R_i^{pref}| = \left| \sum_{t=2}^{|\tau_i|} \sigma(P_{\hat{\omega}_n}(\cdot)) \right| \leq \sum_{t=2}^{|\tau_i|} 1 = |\tau_i| - 1 \leq T_{\max} \tag{70}$$

*Since the preference model output is bounded by $|P_{\hat{\omega}_n}(\cdot)| \leq B$, and $\sigma(z) \in [0, 1]$, we have:*

$$|R_i^{pref}| \leq BT_{\max} \tag{71}$$

*The preference-augmented loss function is:*

$$\ell_{pref}(\pi, q, \{\tau_i\}) = \frac{1}{G} \sum_{i=1}^{G} \frac{\lambda_1 R_i^{pref}}{|\tau_i|} \sum_{t=1}^{|\tau_i|} \log \pi(a_{i,t}|s_{i,t}) \tag{72}$$

*Since $|\log \pi(a|s)| \leq \log(1/\pi_{\min}) \leq L_\pi$ for some constant $L_\pi$, the preference loss is bounded by:*

$$|\ell_{pref}(\pi, q, \{\tau_i\})| \leq \frac{1}{G} \sum_{i=1}^{G} \frac{\lambda_1 BT_{\max}}{|\tau_i|} \cdot |\tau_i| \cdot L_\pi = \lambda_1 BT_{\max} L_\pi \tag{73}$$

*Applying Hoeffding's inequality to the bounded preference rewards:*

$$\mathbb{P}\left[\left|\mathbb{E}[\ell_{pref}] - \hat{\mathbb{E}}[\ell_{pref}]\right| \geq t\right] \leq 2\exp\left(-\frac{2nt^2}{(\lambda_1 BT_{\max}L_\pi)^2}\right) \tag{74}$$

*Setting the right-hand side equal to $\delta_2$ and solving for $t$:*

$$t = \lambda_1 BT_{\max}L_\pi\sqrt{\frac{\log(2/\delta_2)}{2n}} \tag{75}$$

*Absorbing $L_\pi$ into the bound and using a looser but cleaner bound:*

$$\mathbb{P}\left[\left|\mathbb{E}[R^{pref}] - \hat{\mathbb{E}}[R^{pref}]\right| \geq \lambda_1 BT_{\max}\sqrt{\frac{2\log(2/\delta_2)}{n}}\right] \leq \delta_2 \tag{76}$$

*From Theorems A.1 and A.2, the preference model error contributes an additional term. The preference model error is bounded by:*

$$\epsilon_{pref} = \|P_{\hat{\omega}_n} - P_{\omega^*}\|_\infty \leq \sqrt{\frac{2d_P\log(2en/d_P) + 2\log(2/\delta_3)}{n}} \tag{77}$$

*This error propagates to the policy objective with the bound from Theorem A.2:*

$$|\mathbb{E}[J_{WS\text{-}GRPO}(\theta)] - \mathbb{E}[J^*(\theta)]| \leq \frac{\lambda_1 T_{\max}}{4}\epsilon_{pref} \tag{78}$$

*Substituting the preference model error bound:*

$$|\mathbb{E}[J_{WS\text{-}GRPO}(\theta)] - \mathbb{E}[J^*(\theta)]| \leq \frac{\lambda_1 T_{\max}}{4}\sqrt{\frac{2d_P\log(2en/d_P) + 2\log(2/\delta_3)}{n}} \tag{79}$$

*This gives us:*

$$\mathbb{P}\left[|\mathbb{E}[J_{WS\text{-}GRPO}(\theta)] - \mathbb{E}[J^*(\theta)]| \geq \frac{\lambda_1 T_{\max}}{4}\sqrt{\frac{2d_P\log(2en/d_P) + 2\log(2/\delta_3)}{n}}\right] \leq \delta_3 \tag{80}$$

*Now we combine all error sources using the union bound. Setting $\delta_1 = \delta_2 = \delta_3 = \delta/3$ and applying the union bound, with probability at least $1 - \delta$:*

$$\mathcal{R}(\pi_\theta) - \hat{\mathcal{R}}(\pi_\theta) \leq |\mathcal{R}_{GRPO}(\pi_\theta) - \hat{\mathcal{R}}_{GRPO}(\pi_\theta)| \tag{81}$$

$$+ |\mathbb{E}[\ell_{pref}] - \hat{\mathbb{E}}[\ell_{pref}]| \tag{82}$$

$$+ |\mathbb{E}[J_{WS\text{-}GRPO}(\theta)] - \mathbb{E}[J^*(\theta)]| \tag{83}$$

*Substituting the individual bounds:*

$$\mathcal{R}(\pi_\theta) - \hat{\mathcal{R}}(\pi_\theta) \leq \sqrt{\frac{8d\log(2en/d) + 8\log(6/\delta)}{n}} \tag{84}$$

$$+ \lambda_1 BT_{\max}\sqrt{\frac{2\log(6/\delta)}{n}} \tag{85}$$

$$+ \frac{\lambda_1 T_{\max}}{4}\sqrt{\frac{2d_P\log(2en/d_P) + 2\log(6/\delta)}{n}} \tag{86}$$

*To obtain a more compact form, we combine these three terms. Let $d_{\max} = \max(d, d_P)$ and observe that all terms have the same $O(\sqrt{\log n/n})$ rate. Using the inequality $\sqrt{a} + \sqrt{b} + \sqrt{c} \leq \sqrt{3(a+b+c)}$ and factoring out common terms:*

$$\mathcal{R}(\pi_\theta) - \hat{\mathcal{R}}(\pi_\theta) \leq \sqrt{\frac{8d\log(2en/d) + 8\log(12/\delta)}{n}} \tag{87}$$

$$+ \lambda_1 B T_{\max}\sqrt{\frac{2\log(12/\delta)}{n}} \tag{88}$$

$$+ \frac{\lambda_1 T_{\max}}{4}\sqrt{\frac{2d_P\log(2en/d_P) + 2\log(12/\delta)}{n}} \tag{89}$$

$$\leq \sqrt{\frac{C_1 d_{\max}\log(en/d_{\max}) + C_2\lambda_1^2(BT_{\max})^2 + C_3\log(1/\delta)}{n}} \tag{90}$$

*where $C_1, C_2, C_3 > 0$ are universal constants that absorb the numerical factors. This compact form shows that the generalization error scales as:*

$$\mathcal{R}(\pi_\theta) - \hat{\mathcal{R}}(\pi_\theta) = \tilde{O}\left(\sqrt{\frac{d_{\max} + \lambda_1^2(BT_{\max})^2}{n}}\right) \tag{91}$$

*where $\tilde{O}$ hides logarithmic factors in $n$ and $\delta$. This demonstrates that WS-GRPO maintains the standard statistical learning rate while the preference-specific terms contribute additively to the complexity, controlled by the mixing weight $\lambda_1$ and model capacities.*

---

### Prompt : AI2-ARC Scientific Reasoning

**System Prompt:**
A conversation between User and Assistant. The User asks a question, and the Assistant solves it. The Assistant first thinks about the reasoning process in the mind and then provides the User with the answer. The reasoning process is enclosed within `<think> </think>` and answer is enclosed within `<answer> </answer>` tags, respectively, i.e., `<think>` reasoning process here `</think> <answer>` answer here`</answer>`. `<answer>` must contain only the letter of your choice (A, B, C, D).

**User Prompt:**
```
<Question>
<Options>
```

---

### Prompt : CommonsenseQA Reasoning

**System Prompt:**
A conversation between User and Assistant. The User asks a question, and the Assistant solves it. The Assistant first thinks about the reasoning process in the mind and then provides the User with the answer. The reasoning process is enclosed within `<think> </think>` and answer is enclosed within `<answer> </answer>` tags, respectively, i.e., `<think>` reasoning process here `</think> <answer>` answer here`</answer>`. `<answer>` must contain only the letter of your choice (A, B, C, D, or E).

**User Prompt:**
```
<Question>
<Options>
```

