# OpenReview forum: "WS-GRPO: Weakly-Supervised Group-Relative Policy Optimization"
_ICLR.cc/2026/Conference — ICLR 2026 Conference Withdrawn Submission_

### Official Review · Reviewer_5wrv · 2025-10-14

**Soundness:** 1
**Presentation:** 2
**Contribution:** 2
**Rating:** 2
**Confidence:** 5

**Summary:**

GRPO performs well for multi-step reasoning but requires dense, step-wise rewards, which are costly to obtain. In practice, GRPO often relies only on sparse outcome labels, such as final correctness. The authors propose WS-GRPO, a method designed to address this limitation by extracting dense, step-level feedback from weak supervision through a preference learning scheme that distinguishes binary outcomes for trajectories while preserving GRPO’s inherent advantages. The authors present two theoretical analyses of their method and conduct experiments on two base model families and two factuality-based benchmarks to evaluate its effectiveness.

**Strengths:**

- The idea of using stepwise signals to transform sparse rewards into dense ones is interesting and timely. This approach is particularly valuable for critic-free GRPO applications, where learning signals may vanish due to all-positive or all-negative groups.

- The overall structure of the work is solid, with clear motivation and some theoretical analysis supporting the proposed method.

**Weaknesses:**

See below, I provide some suggestions for further improvement:

- **[W1]** The overall manuscript is not clearly written; formatting issues and a lack of writing clarity significantly hinder readability. These issues should be addressed in the next revision. Specifically, please replace all instances of \cite with \citep or \citet as appropriate to ensure correct citation formatting. Additionally, please fill in all missing content, e.g., in line 592: *"Following (?Bartlett & Mendelson, 2002), …"*, to maintain completeness.

---

The effect and rationale for using the preference model are unclear. See detailed comments below:

- **[W2]** The use of a small MLP with only 512 units to learn preferences is not convincing in terms of its effectiveness and generalizability. The authors do not provide training curves or results showing the model’s reward performance on the validation set.

- **[W3]** It is unclear how stepwise reasoning quality is captured through the labeled dataset, as the labels are still based on final outcome correctness. The binary preference learning process continues to rely on outcome-level rewards, raising concerns about the rationale for using such a method to reflect reasoning quality. (Also see [W5].)

- **[W4]** The necessity of using a "weakly supervised" method is not well justified. Why not directly apply DPO to learn preference pairs? No baselines are compared against this approach, making it difficult to assess the claimed advantages. Please clarify why your preference update scheme should be considered more effective than existing alternatives.

- **[W5]** There are no validation or ablation results to support that the method accurately captures stepwise reasoning quality. Specifically, there is no quantitative analysis to substantiate the claim that *"semantic representations of reasoning chains contain implicit quality signals"* (lines 256–257).

---

There are also several limitations and inaccuracies in the theoretical analysis, and it is difficult to see how the presented theory connects to stepwise reasoning quality:

* **[W6]** From a high-level perspective, it is unclear how the theoretical objective reflects *stepwise* accuracy. The pairs $(\tau_i^+, \tau_i^-)$, according to your definition, are still trajectory-level and outcome-based. Therefore, the effectiveness of the preference model cannot be theoretically justified as capturing stepwise quality.

* **[W7]** Even if both bounds are technically correct, their practical significance is unclear. The paper should include a dedicated discussion explaining the empirical meaning of the variables appearing in the bounds and, ideally, provide a rough estimation of their scale in real experiments.

* **[W8]** In Eq. 13, you assume unbiasedness to claim that the bias term has zero mean. This appears to be the key link between outcome-based labels and stepwise quality. However, if my understanding is correct, this assumption is extremely strong and generally unrealistic for reasoning rollouts. It effectively assumes that *"good results always come from good reasoning and bad results always come from bad reasoning"*, which is arbitrary and contradicts the stated goal of distinguishing rollouts at the *stepwise* level. Since this assumption is used, not derived, to eliminate $R_{\text{bias}}$ and obtain later results, the overall proof structure and claims relying on it should be revised.

* **[W9]** The derivation from Eq. 25 to Eq. 28 is unclear. The argument appears to jump from a VC-bound on a loss function to an $\ell_\infty$ deviation bound for real-valued outputs passed through a sigmoid. This step is invalid because VC theory applies to binary classification, not directly to real-valued function classes.

* **[W10]** In Eq. 24, you cannot use a uniform strong convexity constant for unbounded logits, as the binary cross-entropy loss is not globally strongly convex. You must either explicitly assume bounded logits or revise all related parts of the proof to address this issue.

* **[W11]** In the RL theory section, clipping and KL regularization appear to be omitted from the analysis. If these factors are intentionally excluded, please state this clearly; otherwise, they should be incorporated into the theoretical framework for consistency with GRPO.

---

Limitation in GRPO experiment part:
- **[W12]** For the benchmark in table 2, is the metric performance with $\uparrow$ or $\downarrow$? Why WS-GRPO performed even with lower accuracy than Dr. GRPO and GRPO on 75\% of the benchmark?
- **[W13]** Baseline performance before training isn't reported, i.e., it's unclear how the training in GRPO is effective on this benchmark and to assess the relative change?
- **[W14]** The scope of experiment is narrow, as it's only conducted on two factual-based benchmarks; readers will be more interesting to see how your stepwise method could facilitate the tighter step-to-step thinking, e.g., Math benchmarks. Considering to add more experiments in math related area would strengthen your claim.

**Questions:**

See Weakness part. Please address the concerns and questions in the weakness section.

---

### Official Review · Reviewer_vSVH · 2025-11-01

**Soundness:** 2
**Presentation:** 2
**Contribution:** 2
**Rating:** 4
**Confidence:** 4

**Summary:**

The paper proposes WS-GRPO, a two-phase framework that enables group-relative policy optimization under weak supervision. Phase 1 trains a preference model using only trajectory-level outcomes. Phase 2 converts the learned preference signal into dense step-wise rewards by comparing consecutive partial trajectories, and mixes them with terminal rewards inside the GRPO objective. The authors also present a theory: preference consistency, policy robustness to preference error, and a generalization bound decomposing errors from both policy learning and preference modeling. Experiments on AI2-ARC and CommonsenseQA show “competitive” results and architecture sensitivity.

**Strengths:**

**Practical motivation**: turns sparse outcome labels into dense per-step feedback without costly process annotations, it’s useful when step-level PRM labels are unavailable.

**Clear mechanism**: the “adjacent-prefix comparison” is implementable and easy to understand, which integrates cleanly with GRPO’s group-relative advantage.

**Theoretical guarantees**: the paper provides preference-model consistency, linear-type robustness of the policy to preference errors, and a generalization bound, which together help justify moving from weak to dense supervision.

**Problem fit**: the method directly targets a known pain point of GRPO; it benefits from dense process signals, while real data often provide only terminal outcomes.

**Weaknesses:**

**Incremental novelty**: Pairing process rewards or PRMs with GRPO predates this work. For example, TP-GRPO[1] integrates a generative process reward model with GRPO, thereby injecting dense step-wise rewards into GRPO, and introduces a thought-level scheme and an off-policy pipeline.

**Lack of experiments/models**: The paper evaluates only two reasoning datasets, which limits generality. Please broaden coverage to tasks that stress different skills: math (e.g., GSM8K, MATH, OlymMATH), tool-use (e.g., StableToolBench, ToolSANDBOX), and long-chain proofs. Model scope is also narrow, since experiments use only a 3B model. Add smaller and larger scales to show scalability and stability, for example, 1.5B and 7B from the Qwen2.5 family, and consider the latest Qwen3 variants where applicable.

**Wording strength**: The abstract says GRPO “critically depends on dense step-wise rewards.” Literature supports “performs best / benefits from dense signals” rather than strict dependence; the claim should be softened or supported with broader evidence.

**Hyperparameters**: Key hyperparameters are unspecified, and there is no ablation. Please report at least: (i) the preference-model encoder and head, (ii) the mixing weight between step-wise and terminal rewards, and (iii) GRPO group size and KL coefficient, then run small ablations on these. Without this, results are hard to reproduce and may reflect tuning rather than the method, which deep RL studies and reproducibility checklists caution about.

**Questions:**

**Step segmentation**: How are “steps” defined in practice, lines or tokens? If you use textual heuristics, how robust are the results to the segmentation rule? Please provide a short ablation.

**Trajectory-pair construction**: How do you form positive–negative trajectory pairs: number of samples K per prompt, handling ties (both correct or both wrong), duplicate removal, and class balancing across prompts? Please give some examples.

---

### Official Review · Reviewer_WXWE · 2025-11-01

**Soundness:** 2
**Presentation:** 3
**Contribution:** 2
**Rating:** 2
**Confidence:** 3

**Summary:**

This paper introduces WS-GRPO, a weakly-supervised extension of Group-Relative Policy Optimization. WS-GRPO trains a trajectory-preference model from sparse outcome labels, then treats consecutive partial trajectories as preference pairs to synthesize step-level rewards inside GRPO’s normalized advantage framework. The authors also provide theoretical analysis for WS-GRPO. On AI2-ARC and CommonsenseQA, WS-GRPO matches GRPO with zero step annotations, but performance is model/task-sensitive.

**Strengths:**

1. This work frames step-level reward dependency as a weak-supervised problem, and uses a preference model to get step-wise signals without step labels.

2. The writing is clear: a running example (Figure 1) visualizes how partial-trajectory pairs generate step rewards. The notations are consistent and easy to follow.

3. The paper shows that trajectory-level outcome labels contain enough signal for consistent step-wise credit assignment.

**Weaknesses:**

1. Empirical coverage is too narrow:

    (1) Only two datasets (ARC, CSQA) and only 3B-scale models are tested; no results on larger models or math reasoning benchmarks where GRPO is also popular.
    (2) Missing ablation on G (number of group roll-outs): GRPO’s variance reduction relies on large G; with G=8 the baseline itself may be under-powered, inflating WS-GRPO’s apparent competitiveness.
    (3) No comparison with dense PRM baselines (e.g., GRPO+PRM) — the paper claims to “approach” dense supervision but never measures the performance gap against the true upper-bound.

2. Step-level credit assignment remains superficial:

    Equation (6) simply sums per-step preferences and then averages over length, so every step within a trajectory receives identical advantage, which is not fine-grained credit assignment.

3. Sensitivity to preference quality is under-analyzed

   The preference model trained from sparse outcome labels may not perform well for step-level reward. The paper provides no analysis on failure cases: when preference model ranks wrong trajectories higher, WS-GRPO will wrongly amplifies their probability.

4. Hyper-parameter fragility:

    This work suppose fixed $\lambda_1=1, \lambda_2=5$ across both tasks and both models. No schedule or adjustments of parameters are explored.

**Questions:**

Q1: Can you provide some results on math benchmarks (e.g., MATH) and on 7B-scale models? These are the domains where GRPO is widely applied.

   Q2: Pleasw include a dense-PRM baseline (e.g., GRPO + a small supervised PRM) so we can see how close WS-GRPO actually gets to the upper-bound as you mention in the paper.

   Q3: Please report the accuracy of the preference model and correlate it with downstream policy performance.

---

### Note · Authors · 2025-11-23

**Comment:**

We sincerely thank the reviewers for their valuable feedback. To properly address the suggestions, we have decided to withdraw the paper at this stage. We appreciate the time and effort put into the reviews and will work to improve the manuscript.

**Withdrawal Confirmation:**

I have read and agree with the venue's withdrawal policy on behalf of myself and my co-authors.